# Outliers (typically) cannot cause type I errors in one-sample/paired t-tests

**Alan Wisler** ©*

Department of Mathematics and Statistics, Utah State University, Logan, Utah, United States of America

* alan.wisler@usu.edu

## Abstract

The presence of outlying data points can have a significant impact on statistical modeling and significance testing. In the specific context of one-sample t-tests, prior studies have shown (primarily through simulations) that outliers make it more likely for t-tests to fail to reject the null hypothesis. In this study, we investigate the opposite scenario: when an outlier can cause the rejection of the null hypothesis. While it may seem intuitive that outliers aligned with the direction of an effect strengthen that effect, prior studies have shown that this is not always the case. Towards this end, we introduce mathematical bounds on how large outliers can be while still increasing the t-statistic in a given sample. These bounds are validated and supported using Monte-Carlo simulations and a survey of available data sets. From these results, we find that although it is not impossible for outliers to cause significant results in paired or one-sample t-tests, it can only occur under rather narrow circumstances. Specifically, it requires a concordant outlier, a minimal sample size of ($n \geq 15$), and a sufficiently small effect size ($\hat{\mu}/\hat{\sigma} \leq 1/2$). Based on these findings, we argue that the risk of isolated outliers causing type I errors is low in many practical situations, especially when sample sizes are small.

## 1 Introduction

The challenge of dealing with outliers is a pervasive methodological challenge which spans a range of scientific disciplines [1–5]. Although somewhat nebulous, the term *outlier* broadly refers to a sample of data that markedly differs from other members of its group. Despite often cast as *bad* data, outliers are not necessarily the result of experimental error and instead can simply be "an extreme manifestation of the random variability inherent in the data" [6, p. 1]. Regardless of the cause, outliers can disproportionately influence statistical estimates and bias statistical tests, and thus should be handled thoughtfully [7,8]. The best methods for handling outliers are a topic of significant debate across scientific disciplines. A common approach is to simply remove extreme values [9,10], though the criteria for identifying outliers can vary depending on the nature of the data. For a standard homogeneous sample, simple rules such as removing observations more than two or three standard deviations

**Data availability statement:** No real data is used in this manuscript (simulation code available at https://osf.io/yfju9/).

**Funding:** The author(s) received no specific funding for this work.

**Competing interests:** The authors have declared that no competing interests exist.

away from the mean can be employed [11]. However, in data sampled from different populations (or experimental conditions), more complicated questions arise as to whether the identification of outliers should be performed within each condition or across the entire dataset. For example, while [12] points out several issues with identifying outliers separately within conditions, [13] shows that identifying outliers across the entire dataset can also introduce problems. Note that all of these considerations become significantly more complicated in settings where data is multivariate [14,15] or the product of complex dynamic processes [16]. Because of the challenges associated with identifying outliers, and the risk they pose to many common tests, a range of statistical methods have been proposed with the aim of being robust to their presence [17–22]. The aim of this work is not to determine the best method of handling outliers, but to examine a very common statistical test (the paired/1-sample t-test) which is known to be somewhat sensitive to outliers, and to form a better understanding of the effect they can have on it.

In the standard null hypothesis significance testing framework, there are two types of errors that can occur. Type I errors refer to rejecting the null hypothesis when it is true (also known as false positives or false discoveries), and type II errors refer to failing to reject the null hypothesis when it is false (also called false negatives). A number of studies have examined the effects of outliers on common statistical tests and consistently found that their presence increases the risk of type II errors, whereas excluding outliers tends to increase the rate of type I errors [12,13,23,24]. In the case of the basic one-sample or paired Student's t-test, the reason for this is quite clear: the presence of observations that deviate dramatically from the mean increases the sample standard deviation. Since the test statistic is calculated as the ratio of the mean to its associated standard error, an increase in variability inflates the denominator, thereby reducing the t-value and making it less likely that the test will reject the null hypothesis. Nevertheless, this seemingly obvious reasoning bears further scrutiny in certain contexts.

Imagine a study in which ten participants are each tested under two separate conditions, A and B. Suppose the first nine participants all perform slightly better in condition B than in condition A. If the tenth participant were found to perform dramatically better in condition B than condition A, it would be natural to assume that this new data point would only strengthen our confidence in the superiority of condition B over condition A. However, in practice, a paradox arises: if this tenth observation deviates too much from the rest, a paired t-test may be less likely to reject the null hypothesis (and suggest condition B is superior) with the outlier than without it. This counterintuitive effect—that concordant outliers can reduce the likelihood of rejecting the null hypothesis, even when they appear to support the main effect—has already been documented [25]. In a paired t-test, the test statistic is calculated using the mean of the within-subject difference scores, divided by the standard deviation of those differences. A concordant outlier increases both the difference in means across conditions and the standard deviation of the difference scores, and the resulting impact on the t-value depends on how these two changes interact. In some cases, the inflated variability dominates, shrinking the test statistic and reducing statistical power and in others the increase to the mean difference dominates and the t-statistic increases.

However, the precise conditions under which outliers increase or decrease the t-statistic, and the corresponding likelihood of rejecting the null hypothesis, remain poorly understood.

Whereas much of the previous literature highlights the risk of outliers inflating type-II errors in t-tests, this work focuses on the risk they pose for type-I errors (or false positives). Specifically, we aim to develop a clearer understanding of the exact conditions under which outlying values can *cause* a paired t-test to reject the null hypothesis. For our purposes, we define an outlier as causing the rejection of the null if: 1) the null hypothesis is rejected when the outlier is included, and 2) the null hypothesis would not be rejected if the outlier were removed or replaced with any non-outlying value. As the definition for outlier is highly subjective, we explore the introduction of points between 0-10 standard deviations away from the mean and leave it to readers to decide which points meet their definition, although we will highlight some specific implications for 2-$\sigma$ and 3-$\sigma$ outlier thresholds. Within this framework, we use both mathematical derivations and empirical simulations to illustrate how different magnitudes of "outlier" can influence the t-statistic and corresponding statistical inferences. Finally, although the framework used in this paper is defined in terms of a generic one-sample t-test, we can easily think of these samples as differences and generalize these findings to paired t-tests as well. Thus, we will largely use the terms "one-sample" and "paired" t-test interchangeably.

The organization of this paper is laid out as follows. Sect 2 introduces the mathematical framework for this paper and presents two upper bounds on how large an outlier can be while still increasing the t-statistic. Sect 3 then presents three experiments designed to validate these bounds and develop a more concrete understanding of the scenarios under which concordant outliers can and cannot cause type I errors. The first two experiments use simulations to assess the effects of concordant outliers under different sample sizes and effect sizes. The third experiment looks at a set of fifty real-world datasets to see how frequently concordant outliers are present and whether or not they are driving any of the significant results in these data. Finally, Sect 4 presents a discussion of our findings to synthesize the main takeaways and relate these findings to the existing literature on this topic.

## 2 Framework and derivations

The methods for this paper will be laid out as follows. We will first introduce the basic framework and notation that will be used in this paper to describe a scenario in which an arbitrary set of data is influenced by the introduction of a new observation that is greater than the mean of the existing data. In general, we observe that when the new observation is only slightly greater than the mean its introduction will generally increase the t-statistic. However, when the new observation is several standard deviations above the mean it will often decrease the t-statistic. Thus, the second part of our methods derives closed form expressions for how many standard deviations above the mean the new observation should be in order to 1) increase the test statistic and 2) maximally increase the test statistic.

### 2.1 Problem description

Let us start by defining a set of data $\mathbf{x} = [x_1, x_2, \ldots, x_n]$ with sample mean and variance:

$$\hat{\mu} = \frac{1}{n} \sum_{i=1}^{n} x_i, \quad \hat{\sigma}_x^2 = \frac{1}{n-1} \sum_{i=1}^{n} (x_i - \hat{\mu})^2. \tag{1}$$

Now we extend the dataset with a new observation $\mathbf{x}' = [x_1, x_2, \ldots, x_n, x_{n+1}]$, where

$$x_{n+1} = \hat{\mu} + \Delta. \tag{2}$$

For our analysis, the new observation $x_{n+1}$ serves the role of a concordant outlying data point that is added to an existing sample. Note that we will consider a wide range of $\Delta$ values in our analysis and describing $x_{n+1}$ as an outlier when $\Delta$

is small might be misleading. To avoid drawing an arbitrary line on what is and is not considered an outlier in our analysis, we keep this definition overly broad and leave the task of disambiguating which cases accurately describe outliers to our discussion.

The fundamental question we seek to address in this paper is whether or not it is possible for the introduction of an outlying point into the data to result in false discoveries for 1-sample student t-tests. More specifically, we ask the question: under what conditions of sample size ($n$), sample mean $\hat{\mu}$, and perturbation magnitude $\Delta$, does the introduction of observation $x_{n+1}$ increase the t-statistic. Without loss of generality, we assume that:

1. $\hat{\sigma}_x^2 = 1$
2. $\hat{\mu} \geq 0$ and $\Delta \geq 0$

While the first condition appears restrictive it merely assumes that the data has been scaled to unit variance prior to conducting the analysis. For example, suppose we have sample $\tilde{\mathbf{x}}$ with mean $\hat{\mu}_{\tilde{\mathbf{x}}}$ and variance $\hat{\sigma}_{\tilde{\mathbf{x}}} \neq 1$. We could then define $\mathbf{x} = \tilde{\mathbf{x}}/\hat{\sigma}_{\tilde{\mathbf{x}}}$, and our analysis would then hold for the scaled data. Importantly, this scaling has no effect on the t-statistic since it affects the numerator and denominator equally. This condition also makes interpretation of our results significantly easier since parameters like $\hat{\mu}$ and $\Delta$ are now measured as number of standard deviations, rather than along the context-specific scale of the original data. So, if our criteria for outliers were defined as any points more than three standard deviations away from the mean, that would reflect cases where $|\Delta| > 3$. Additionally, because $\hat{\sigma}_x^2 = 1$ we can think of the sample mean as representing the effect size in our data (since $\hat{\mu} = \hat{\mu}/\hat{\sigma}_x^2$), and will frequently refer to it as such going forward.

Regarding the second condition, since our results will be based on the relative magnitude of the t-statistic for the modified data, flipping the sign of both $\hat{\mu}$ and $\Delta$ would not affect our findings (since the t-statistic depends on the magnitude of the effect, not its direction). Thus, this condition only limits generality in the sense that it requires the added outlying value to be in the same direction as the observed effect. This omission represents a rather uninteresting case, since it is generally assumed that outliers in the opposite direction of the observed effect will not result in false discoveries.

Using (1), we can derive the sample mean and variance of the modified data as (full derivation in S1 File):

$$\hat{\mu}' = \hat{\mu} + \frac{\Delta}{n+1}, \hat{\sigma}_{x'}^2 = \frac{(n-1)}{n} + \frac{\Delta^2}{n+1} \qquad (3)$$

and from this the t-statistic of the original and modified data are

$$t = \frac{\hat{\mu}\sqrt{n}}{\hat{\sigma}_x} = \hat{\mu}\sqrt{n} \qquad (4)$$

and

$$t' = \frac{\hat{\mu}'\sqrt{n+1}}{\hat{\sigma}_{x'}} = \frac{\sqrt{n+1}(\hat{\mu} + \frac{\Delta}{n+1})}{\sqrt{\frac{(n-1)}{n} + \frac{\Delta^2}{n+1}}}. \qquad (5)$$

One interesting observation we can make from these equations is that the both $t$ and $t'$ are dependent on only three parameters: the sample mean of the original data ($\hat{\mu}$), the sample standard deviation of the original data ($\hat{\sigma}_x$), and the number of standard deviations the new sample is placed above the mean ($\Delta$). This means that the effect of an outlier (or any new observation) on the t-statistic will be the same for any two datasets of equal mean and variance regardless of how they are distributed. This is an important point which will (in the subsequent analysis) allow us to make highly general claims about how outliers affect the t-statistic without relying on distributional assumptions.

## 2.2 Deriving bounds on outlier magnitude

Here we introduce theoretical limits on $\Delta$ that provide guarantees on how the new observation will affect the results of a one-sample t-test. The first observation we would like to highlight is that as the value of the new observation gets arbitrarily large ($\Delta \rightarrow \infty$) the test statistic goes to one ($t' \rightarrow 1$). For a two-tailed t-test, this yields a p-value of $p \approx 0.317$. Thus, given a sufficiently large outlier, a t-test will always yield a null result regardless of the strength of evidence in the original (outlier-free) data. Note that these asymptotic characteristics have already been shown in [25]. The second observation is that there is an upper bound on $\Delta$ (which we will call $\Delta_1^*$), defined in terms of only $\hat{\mu}$ and $n$, beyond which the introduction of $x_{n+1}$ will reduce the t-statistic of the original data. This upper bound is expressed formally in Theorem 2.2 below.

**Theorem 1.** *If $\hat{\mu} > \frac{1}{\sqrt{n}}$ and $\Delta \geq \Delta_1^*$ where*

$$\Delta_1^* = \frac{\hat{\mu}\left(n + 1 + \sqrt{2\hat{\mu}^2 n^2 + 2\hat{\mu}^2 n + n^2 - 1}\right)}{n\hat{\mu}^2 - 1}$$

*then $t' \leq t$.*

The proof for Theorem 2.2 is provided in S2 File. The third and final observation is that there is a second upper bound on $\Delta$ (which we will call $\Delta_2^*$) defined in terms of only $\hat{\mu}$ and $n$, beyond which perturbing $x_{n+1}$ any further from the mean will reduce $t'$. This upper bound is expressed formally in Theorem 2.2 below.

**Theorem 2.** *If $\Delta \geq \Delta_2^*$ where*

$$\Delta_2^* = \frac{n - 1}{n\hat{\mu}}$$

*then $\frac{dt'}{d\Delta} \leq 0$.*

The proof for Theorem 2.2 is provided in S3 File. It is also fairly straightforward to show that $\Delta_1^* > \Delta_2^*$ under the predefined conditions (also included in S3 File). These two theorems provide general insights into the maximum values a new outlier can take while still exerting positive effects on the t-statistic. Relating this back to our theoretical framework $\Delta$ describes the number of standard deviations above the mean where the new observation ($x_{n+1}$) is located. The value of $\Delta_2^*$ depicts the exact location where the new observation will maximally increase the t-statistic. Thus, if we imagine we can control the placement of this new observation, the further we move it away from that location the less it will increase the t-statistic. And if we move it far enough away from this point in the positive direction, $\Delta_1^*$ describes the point at which its inclusion begins to lower the t-statistic (relative to its absence). Taken together with the asymptotic characteristics of $\Delta$, this will eventually lead to failure to reject the null hypothesis. Based on the provided definitions for when an outlier *causes* rejection of the null hypothesis, $\Delta_2^*$ provides a relatively clear indication of when this can occur. For any sample size ($n$) and sample mean ($\hat{\mu}$) we consider, if the value of $\Delta_2^*$ would not meet our definition of outlier then no outlier can cause the rejection of the null, since no outlying observation can increase the t-statistic more than $x_{n+1} = \hat{\mu} + \Delta_2^*$. For example, if $n = 10$ and $\hat{\mu} = 1.0$, then $\Delta_2^* = 0.90$, and any new observations added to the data cannot increase the t-statistic more than one exactly 0.9 standard deviations above the mean. Thus, where the addition of some outlying points might increase the t-statistic in this scenario its influence would necessarily be less than some alternative non-outlying sample.

To better understand what these results tell us about the relationship between the placement of the new data point $x_{n+1}$ and $t'$, Fig 1 displays the curves of (a) the new t-statistic ($t'$) and (b) the empirical influence function for the t-statistic across $\Delta$ for four different sample means $\hat{\mu} = 0.1, 0.25, 0.5, 1.0$. These plots also display where $\Delta_1^*$ and $\Delta_2^*$ fall on each of these curves. Going in order, the first plot shows $t'$, which starts at different values for the four means, but in each case ascends to a particular peak then begins to decline. The dotted purple line represents the ($\Delta_2^*, t'$) pairs for a continuous

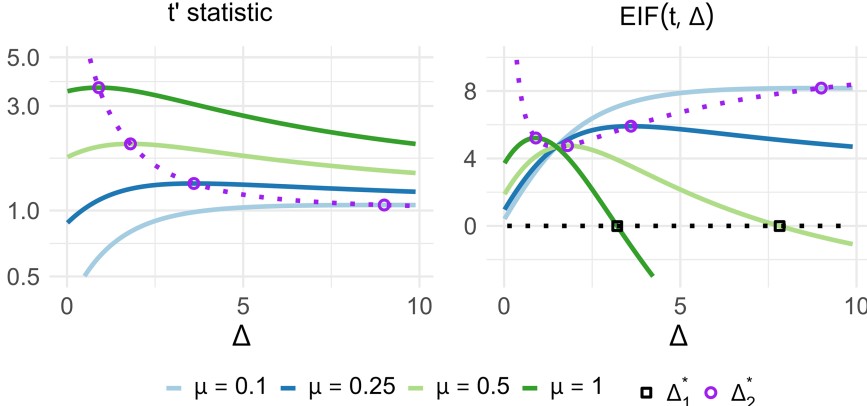

**Fig 1. Curves displaying various measures of the t-statistic relative to the placement of the added sample $x_{n+1}$ which is characterized by $\Delta$ (the number of standard deviations it falls above the mean).** The left plot displays the modified t-statistic ($t'$) and the right plot plot displays the empirical influence function (EIF) of the t-statistic (which is a scaled measure of the difference between the modified and original t-statistic). Where appropriate the derived bounds $\Delta_1^*$ and $\Delta_2^*$ are displayed to show how they intersect with these curves.

range of $\hat{\mu}$ values. As expected, this line intersects with the $t'$ curves at the peaks of each $t'$-curve and these intersections occur at higher $\Delta$ values for lower $\hat{\mu}$ curves. The second plot displays the empirical influence function (EIF) of the t-statistic, which we define as:

$$EIF(t, \Delta) = (n + 1)[t' - t]. \tag{6}$$

This plot displays how much the t-statistic changes as a result of the newly added sample. Note that while the EIF is more typically defined in terms of the location of the added sample, we define it terms of $\Delta$ (the location of the added sample relative to the mean) for consistency with other visualizations. Similar to the first plot, the purple line indicates the peak in the EIF for each curve, which occurs at the same $\Delta$ values as before. For this plot we also see a line for $\Delta_1^*$ at $EIF(t, \Delta) = 0$, indicating the transition point between positive and negative EIF values, or alternatively the point at which the new t-statistic ($t'$) falls below the original t-statistic ($t$).

## 3 Experiments and results

In this section, we introduce three experiments to validate and support the previous derivations. The first experiment will use a Monte-Carlo simulation to empirically verify the accuracy of the bounds introduced in Sect 2, and to quantify what values for sample means ($\hat{\mu}$) and outlier magnitudes ($\Delta$) result in changes to t-test results for three different sample sizes ($n = 10, 25, 100$). The second experiment mirrors the first, but considers all sample sizes between $n = 2$ and $n = 100$ to determine the maximum value of $\Delta_2^*$ that occurs in cases where the significance result becomes significant after the introduction of the new data point. As $\Delta_2^*$ describes the location of the new observation that maximally increases the t-statistic, this value provides an approximate upper bound on the outlier criteria under which an outlier can possibly cause a type I error at a given sample. Finally, experiment 3 applies this methodological framework to a survey of paired datasets to examine whether any cases of outliers causing type I errors can be identified.

### 3.1 Experiment 1: Comparison of bounds

To validate the previously described results we conduct a Monte-Carlo simulation. The Monte-Carlo simulation begins by randomly generating the mean of the original data and the magnitude of the outlier according to the following uniform

distributions: $\mu \sim U(0, 1)$, $\Delta \sim U(0, 10)$. From this, a raw version of the original dataset $\mathbf{x}_r = [x_1, x_2, \ldots, x_n]$ is generated by sampling $n$ values from a Normal distribution with $\mu = 0$ and $\sigma = 1$:

$$x_i \sim N(\mu, 1). \tag{7}$$

This raw data is then normed to create

$$\mathbf{x} = \frac{\mathbf{x}_r}{\hat{\sigma}_r} \tag{8}$$

where $\hat{\sigma}_r$ is the sample standard deviation of the raw data. This ensures that $\hat{\sigma} = 1$ as previously assumed. Note that while the data in this simulation is generated from a normal distribution, normality of the data is not required for our analysis to hold. Since $t$ and $t'$ can be expressed solely in terms of $\hat{\mu}$, $n$, and $\Delta$ the process generating the data has no effect on the results beyond its relation to these parameters. Thus, this experiment could be repeated using any non-normal distribution (with the same sample mean and variance) to achieve the same results. Alternative versions of this experiment demonstrating near identical results for non-normal cases are displayed in S4 File. From here, the modified dataset is defined just as in our model: $\mathbf{x}' = [x_1, x_2, \ldots, x_n, x_{n+1}]$ where $x_{n+1} = \hat{\mu} + \Delta$. As we are interested in not only how the introduction of $x_{n+1}$ affects statistical testing, but also how further perturbation of $x_{n+1}$ would influence statistical outcomes, we also generate a third dataset

$$\mathbf{x}'' = [x_1, x_2, \ldots, x_n, x_{n+1} + \delta] \tag{9}$$

where $\delta = 10^{-4}$ is a very small perturbation constant meant to simulate calculation of a local derivative. Comparing $\mathbf{x}''$ with $\mathbf{x}'$ provides a simple way of determining whether pushing $x_{n+1}$ further from the mean will increase or decrease the t-statistic. For each of the three datasets ($\mathbf{x}$, $\mathbf{x}'$, and $\mathbf{x}''$), we calculate the t-statistics and associated p-values which we will call $t$, $t'$, $t''$ and $p$, $p'$, $p''$ respectively.

From here, within each run of the Monte-Carlo simulation, we will investigate several cases:

- **Case 0** ($t > t'$ and $t' > t''$) : Adding $x_{n+1}$ reduced the t-statistic
- **Case 1** ($t < t'$ and $t' > t''$) : Adding $x_{n+1}$ increased the t-statistic, but less than if $x_{n+1}$ were closer to the mean
- **Case 2** ($t < t'$ and $t' < t''$) : Adding $x_{n+1}$ increased the t-statistic more than any observation closer to the mean would have

If the results in Theorems 1 and 2 are valid and their conditions are met ($\hat{\mu} \geq 1/\sqrt{n}$), it should be the case that any observations with $\Delta \geq \Delta_1^*$ fall into Case 0, any observations with $\Delta_2^* \leq \Delta \leq \Delta_1^*$ fall into Case 1, and any observations with $\Delta \leq \Delta_2^*$ fall into Case 2. The first goal of the simulation is to validate whether the theoretical bounds correctly categorize the empirical t-statistics. The second objective is to examine cases where either the original or modified dataset meets the standard statistical significance threshold $p, p' < 0.05$.

Based on this setup, we run 10,000 iteration Monte-Carlo simulation across three different sample sizes ($n = 10, 25, 100$). Fig 2 displays the results of these simulations via a scatter plot of $\Delta$ vs. $\hat{\mu}$ with lines for the two bounds $\Delta_1^*$ and $\Delta_2^*$. Plots in the top row display individual iterations of the simulation color coded by the previously described cases. These plots clearly illustrate that the bounds accurately delineate these cases for all three sample sizes. As would be expected from the formula, $\Delta_1^*$ is not noticeably affected by the sample size, while the $\Delta_2^*$ line generally shifts leftwards as $n$ gets larger. Plots in the bottom row of Fig 2 display the same results, but this time color coded based on whether $p$ and $p'$ are greater than or less than the significance threshold 0.05. Since the value of $p$ is determined entirely by the values of $\hat{\mu}$ and $n$ (because $\hat{\sigma}$ is fixed), there is a clear separation between points for which $p \geq 0.05$ and $p' < 0.05$ determined by a fixed value of $\hat{\mu}$ for each $n$. We also see that this $\hat{\mu}$-threshold moves left as the sample size increases, and with this leftward shift we see increases in both $\Delta_1^*$ and $\Delta_2^*$. Therefore, for smaller sample sizes there is a lower upper limit on the

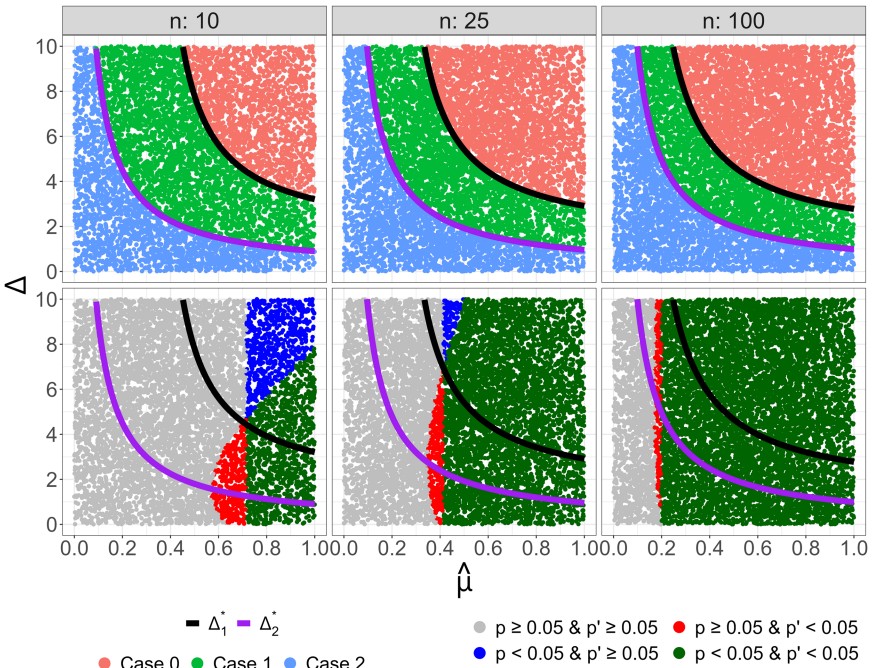

**Fig 2. Simulation results for each sample size** $n \in \{10, 25, 100\}$**, where each dot represents one iteration of the Monte-Carlo simulation.** Each plot is displayed as outlier magnitude ($\Delta$) vs. sample mean ($\hat{\mu}$), with lines depicting the two bounds ($\Delta_1^*, \Delta_2^*$). Points colored by either case (top row), which indicates how the new observation affects the t-statistic for that ($\Delta, \hat{\mu}$) pair, or $p$-values (bottom row), which indicate whether the $p$-value falls below the 0.05 threshold both with and without the new observation. Note that cases in the top row are perfectly separated by the $\Delta_1^*$ and $\Delta_2^*$ lines supporting the validity of the proposed bounds.

magnitude of outliers capable of producing false discoveries. In the $n = 10$ case, for example, we observe $\Delta$ values as high as 4.39 yielding a significant result in the modified data, despite no significance in the original data. However, it is important to point out that in these instances, the added data point could have been much smaller and still produced a significant p-value. If we examine the maximum value of $\Delta_2^*$ in those trials, we see that it never exceeds 1.58. Since these experiments are conducted on normalized data, this means that a new observation need not be more than 1.58 standard deviations above the mean to maximize the t-statistic (and thus minimize the corresponding $p$-value). In the larger samples simulations $n = 25$ and $n = 100$ the maximum observed values for $\Delta_1^*$ in trials where $p \geq 0.05$ and $p' < 0.05$ was 2.77 and 5.83, respectively. Therefore, in larger sample sizes it is possible for outlying samples to meet our definition for causing the rejection of the null hypothesis, although these cases represent a relatively narrow proportion of the observed trials. The following subsection will further explore the relationship between sample size and the maximally influential outlier value $\Delta_2^*$ at the significance threshold.

### 3.2 Experiment 2: Maximum outlier magnitude vs. sample size

In the previous simulation we observed that the values of $\Delta$ likely to change the outcome of the significance test increased for larger sample sizes. This was partially a result of the dependency of $\Delta_1^*$ and $\Delta_2^*$ on $n$, but primarily stems from the inverse relationship between $\Delta_1^*$, $\Delta_2^*$ and $\hat{\mu}$ combined with the fact that the effect size ($\hat{\mu}$) needed for statistical significance decreases for larger $n$. Our goal in this simulation is to develop a better understanding for the maximum value an outlier can take while still causing a statistically significant result in the modified sample. Remember, per our definitions, an outlier causing a significant result requires that 1) its inclusion leads to a significant result and 2) replacing it with any non-outlying value would not lead to a significant result. Per these definitions, it is only possible for outliers to cause false

discoveries when $\Delta_2^*$ meets our outlier criteria as otherwise any outlying sample that yields a significant result could be replaced with a non-outlying sample at $x_{n+1} = \hat{\mu} + \Delta_2^*$ to achieve the same result. Thus the goal of this simulation is to estimate the maximum value of $\Delta_2^*$ in cases where $p \geq 0.05$ and $p' < 0.05$ for a given sample size. To achieve this, we repeat the same 10,000 iteration Monte-Carlo simulation from before at every sample size $n = 2, 3, ..., 100$ and for each sample size measure the maximum value of $\Delta_2^*$ corresponding to trials where $p \geq 0.05$ and $p' < 0.05$. The results of this simulation are presented in Fig 3.

These results offer additional insights into the potential for concordant outliers to cause false discoveries across various sample sizes. If we define outliers as any point that is more than two standard deviations away from the mean, empirical results suggest that it is only possible for outliers to cause rejection of the null hypothesis at sample sizes greater than fourteen. If we impose as stricter definition and require outliers be more than three standard deviations from the mean, the minimum sample size increases to $n \geq 30$ before concordant outliers can cause the null hypothesis to be rejected. Also note that these sample size requirements are necessary for concordant outliers to cause false positives, but not sufficient by themselves. Using the $n = 100$ simulation in Fig 2 as an example, we see that the cases where $p \geq 0.05$ and $p' < 0.05$ still only occur for a very narrow range of $\hat{\mu}$-values.

### 3.3 Experiment 3: Examination of outlier effects in real paired data

In the final experiment, we will look at some real-world datasets containing paired observations to see how frequently concordant outliers occur in these datasets and how they influence paired t-test results. This experiment also provides the opportunity to illustrate how this approach can be applied to real world datasets to characterize the influence of existing data points. To attain a set of different datasets to examine which contain paired observations, we draw from two existing R packages that contain built in repositories of paired datasets, the PairedData package [26], and the BSDA package [27]. Together these libraries provide a total of 43 datasets. As some of these datasets contain more than 1 set of paired observations we then have 50 sets of paired samples for this analysis.

For each dataset, we go through the following procedure. Let us call the set of paired observations loaded from the data $\mathbf{y}_a$ and $\mathbf{y}_b$. We can calculate the original difference pairs as $\bar{\mathbf{x}}' = \mathbf{y}_b - \mathbf{y}_a$. If this doesn't result in $\hat{\mu}_{\bar{\mathbf{x}}'} > 0$, then instead

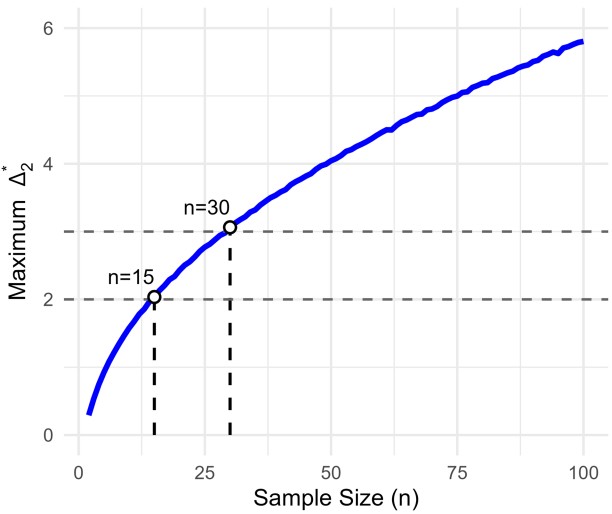

**Fig 3. Outlier magnitude that maximally increases the t-statistic ($\Delta_2^*$) at the significance threshold ($p < 0.05$) as a function of sample size ($n$).** Sample sizes where this value crosses notable 2-$\sigma$ and 3-$\sigma$ outlier thresholds are highlighted to indicate the approximate minimum sample size under which outliers can cause type I errors under each definition.

assign $\tilde{\mathbf{x}}' = \mathbf{y}_a - \mathbf{y}_b$. Now, we identify a candidate concordant outlier as the maximum value in these difference pairs and remove it from the sample.

$$\tilde{\mathbf{x}} = \tilde{\mathbf{x}}' \setminus x^* \tag{10}$$

where $x^* = \max(\tilde{\mathbf{x}}')$. The original data matching our framework ($\mathbf{x}$) is then calculated as the rescaled version of this data

$$\mathbf{x} = \frac{\tilde{\mathbf{x}}}{\hat{\sigma}_{\tilde{\mathbf{x}}}} \tag{11}$$

and the modified data ($\mathbf{x}'$) is created by re-adding the scaled version of the candidate outlier back into this set

$$\mathbf{x}' = \mathbf{x} \cup \frac{x^*}{\hat{\sigma}_{\tilde{\mathbf{x}}}}. \tag{12}$$

Using this data, we calculate the t-statistics both with and without the candidate outlier ($t$ and $t'$ respectively), their associated p-values ($p$ and $p'$). We can also calculate

$$\Delta = \frac{x^* - \hat{\mu}_{\tilde{\mathbf{x}}}}{\hat{\sigma}_{\tilde{\mathbf{x}}}} \tag{13}$$

and

$$\Delta_2^* = \frac{(n-1)\hat{\sigma}_{\tilde{\mathbf{x}}}}{n\hat{\mu}_{\tilde{\mathbf{x}}}} = \frac{n-1}{n\hat{\mu}_{\mathbf{x}}} \tag{14}$$

After going through this process, four of the 50 samples showed negative sample means after the candidate outlier was removed and were excluded from subsequent analysis. The results of this analysis are displayed comprehensively in Table 1. The primary goal of this analysis is to determine how frequently concordant outliers are present in these data and whether instances of them *causing* rejection of the null hypotheses can be identified. Of the 46 paired samples, 23 contained a concordant outlier more than 2 standard deviations above the mean. Of these 23 cases there was only one dataset (the Vocab data) where the results of the test were altered by the concordant outlier. Although this dataset clearly meets the outlier criteria ($\Delta = 3.055$) and the outlier affects the results of the test, this still does not qualify as causing the rejection since $\Delta_2^* = 1.662$ in this instance. Now, if we instead examine cases where the largest concordant observation did not meet the outlier threshold, we see that of those 23 cases in which the maximum observation ($x^*$) fell below the 2-$\sigma$ threshold eight datasets showed a significant result when the candidate outlier was included. Consequently in these datasets it is far more common for a non-outlying observation to yield a rejection that didn't exist before than for and outlying one.

Visualizations of these results are displayed in Fig 4 where each dot represents the results for a single dataset. The first scatter plot displays the updated t-statistic with the maximum concordant observation relative to the t-statistic without it. Although this plot shows the t-statistic increases in the vast majority of cases (37 of 46 according to Table 1) we know from our previous discussion that the majority of these cases the concordant observation increasing the t-statistic does not meet the 2-$\sigma$ outlier threshold. In contrast, all nine of the cases where the t-statistic decreases the concordant observation exceeds this threshold. Note that by definition cases where the concordant observation decreases the t-statistic are cases where $\Delta > \Delta_1^*$. Looking at these nine cases in Table 1 we see that this tends to occur for datasets with larger effect sizes ($\hat{\mu} > 0.8$). This makes sense as $\Delta_1^*$ is inversely related to $\hat{\mu}$. Interestingly, for three of these datasets (Blink, Grain2, and Oxytocin) the $\Delta_1^*$ is less than two, meaning that new observations in these samples don't have to be that much greater than the mean in order to negatively affect the t-statistic. The second set of plots displays the point where the concordant observation will maximally increase the t-statistic ($\Delta_2^*$) relative to the maximum observation ($\Delta$). There are

**Table 1**. **Summary of results for each of the datasets analyzed in Experiment 3.** Note that (in line with the described experiment) all statistics (except $T'$ and $p'$) are calculated after the maximum concordant observation has been removed. Rows in which the p-value falls below the 0.05 threshold when the maximum concordant observation is introduced ($p > 0.05$ & $p' < 0.05$) are highlighted in gray. Note that as Theorem 2.2 requires that $\hat{\mu} > 1/\sqrt{n}$, $\Delta_1^*$ is not reported for datasets where this assumption is violated.

| Data Set | $n$ | $\hat{\mu}$ | $T$ ($p$) | $T'$ ($p'$) | $\Delta$ | $\Delta_1^*$ | $\Delta_2^*$ |
|---|---|---|---|---|---|---|---|
| Anorexia | 16 | 1.004 | 4.017 (0.001) | 4.185 (0.001) | 2.383 | 3.009 | 0.933 |
| Barley | 11 | 1.252 | 4.151 (0.002) | 3.013 (0.012) | 6.231 | 2.706 | 0.726 |
| Blink | 11 | 3.251 | 10.784 (0.000) | 9.739 (0.000) | 2.733 | 1.860 | 0.280 |
| Blink2 | 11 | 0.882 | 2.925 (0.015) | 3.141 (0.009) | 2.581 | 3.506 | 1.031 |
| BloodLead | 32 | 1.044 | 5.908 (0.000) | 5.783 (0.000) | 3.249 | 2.793 | 0.928 |
| ChickWeight | 9 | 0.859 | 2.578 (0.033) | 2.860 (0.019) | 2.455 | 3.743 | 1.035 |
| Corn | 14 | 0.483 | 1.808 (0.094) | 2.178 (0.047) | 1.563 | 6.838 | 1.921 |
| GDO | 17 | 0.250 | 1.033 (0.317) | 1.406 (0.178) | 1.816 | 135.361 | 3.757 |
| GDO | 17 | 0.151 | 0.623 (0.542) | 1.108 (0.283) | 2.784 | - | 6.226 |
| Grain | 8 | 1.608 | 4.547 (0.003) | 4.479 (0.002) | 2.569 | 2.440 | 0.544 |
| Grain2 | 5 | 3.266 | 7.303 (0.002) | 6.708 (0.001) | 2.449 | 1.983 | 0.245 |
| GrapeFruit | 24 | 0.588 | 2.879 (0.008) | 3.157 (0.004) | 1.999 | 4.552 | 1.631 |
| HorseBeginners | 7 | 2.108 | 5.577 (0.001) | 6.190 (0.000) | 1.337 | 2.196 | 0.407 |
| IceSkating | 6 | 0.984 | 2.411 (0.061) | 2.935 (0.026) | 1.578 | 3.638 | 0.847 |
| Iron | 9 | 0.632 | 1.897 (0.094) | 2.327 (0.045) | 1.897 | 5.432 | 1.405 |
| PrisonStress | 10 | 0.703 | 2.224 (0.053) | 2.645 (0.025) | 1.603 | 4.529 | 1.280 |
| PrisonStress | 14 | 0.634 | 2.371 (0.034) | 2.684 (0.018) | 2.391 | 4.671 | 1.465 |
| Rugby | 92 | 0.176 | 1.688 (0.095) | 1.952 (0.054) | 3.694 | 17.864 | 5.619 |
| Sewage | 7 | 1.069 | 2.828 (0.030) | 3.375 (0.012) | 1.269 | 3.248 | 0.802 |
| Shoulder | 14 | 0.185 | 0.694 (0.500) | 1.199 (0.250) | 2.844 | - | 5.007 |
| SkiExperts | 11 | 0.061 | 0.204 (0.843) | 0.639 (0.536) | 1.614 | - | 14.812 |
| Sleep | 9 | 1.886 | 5.659 (0.000) | 4.062 (0.003) | 5.086 | 2.240 | 0.471 |
| Tobacco | 7 | 1.087 | 2.875 (0.028) | 2.625 (0.034) | 4.118 | 3.204 | 0.789 |
| anscombe2 | 14 | 0.469 | 1.755 (0.103) | 2.117 (0.053) | 1.410 | 7.202 | 1.979 |
| anscombe2 | 14 | 0.466 | 1.742 (0.105) | 2.113 (0.053) | 1.581 | 7.299 | 1.994 |
| anscombe2 | 14 | 0.461 | 1.723 (0.109) | 2.104 (0.054) | 1.962 | 7.448 | 2.016 |
| anscombe2 | 14 | 0.464 | 1.738 (0.106) | 2.117 (0.053) | 2.126 | 7.335 | 1.999 |
| Asthmati | 8 | 0.814 | 2.303 (0.055) | 2.753 (0.025) | 1.691 | 4.084 | 1.075 |
| Blood | 14 | 0.079 | 0.295 (0.772) | 0.682 (0.507) | 1.578 | - | 11.767 |
| Cabinets | 19 | 0.576 | 2.510 (0.022) | 2.716 (0.014) | 3.271 | 4.861 | 1.645 |
| Coffee | 8 | 0.227 | 0.642 (0.542) | 1.177 (0.273) | 2.042 | - | 3.857 |
| Corn | 11 | 1.061 | 3.518 (0.006) | 3.831 (0.003) | 1.935 | 3.023 | 0.857 |
| Fitness | 8 | 0.814 | 2.303 (0.055) | 2.753 (0.025) | 1.691 | 4.084 | 1.075 |
| German | 9 | 1.054 | 3.162 (0.013) | 3.638 (0.005) | 1.318 | 3.131 | 0.843 |
| Kinder | 7 | 0.611 | 1.616 (0.157) | 2.117 (0.072) | 1.333 | 6.624 | 1.403 |
| Lowabil | 11 | 0.979 | 3.247 (0.009) | 3.604 (0.004) | 1.790 | 3.213 | 0.929 |
| Movie | 11 | 0.483 | 1.604 (0.140) | 2.000 (0.071) | 1.289 | 7.840 | 1.880 |
| Oxytocin | 10 | 3.175 | 10.040 (0.000) | 8.506 (0.000) | 3.190 | 1.881 | 0.283 |
| Rehab | 19 | 0.086 | 0.375 (0.712) | 0.857 (0.402) | 2.641 | - | 11.002 |
| Selfdefe | 8 | 0.585 | 1.655 (0.142) | 2.121 (0.067) | 1.755 | 6.594 | 1.495 |
| Skin | 10 | 0.802 | 2.537 (0.032) | 2.512 (0.031) | 4.040 | 3.912 | 1.122 |
| Speed | 14 | 0.110 | 0.410 (0.688) | 0.990 (0.339) | 3.268 | - | 8.467 |
| Spelling | 8 | 0.677 | 1.916 (0.097) | 2.286 (0.052) | 2.511 | 5.163 | 1.292 |
| Stress | 11 | 0.665 | 2.206 (0.052) | 2.602 (0.025) | 1.773 | 4.712 | 1.367 |
| Twin | 8 | 0.681 | 1.926 (0.096) | 2.400 (0.043) | 1.414 | 5.126 | 1.285 |
| Vocab | 13 | 0.555 | 2.003 (0.068) | 2.296 (0.039) | 3.055 | 5.672 | 1.662 |

a few interesting observations we can make from this plot. First, in the majority of datasets $\Delta_2^* < 2$, meaning that for these datasets no outlying point can increase the t-statistic more than a point within two standard deviations of the mean. We also observe that in the majority of these cases $\Delta > \Delta_2^*$, which indicates that the test statistics in these data would be greater if the maximum observation were actually smaller. Thus while some of these data show statistically significant test

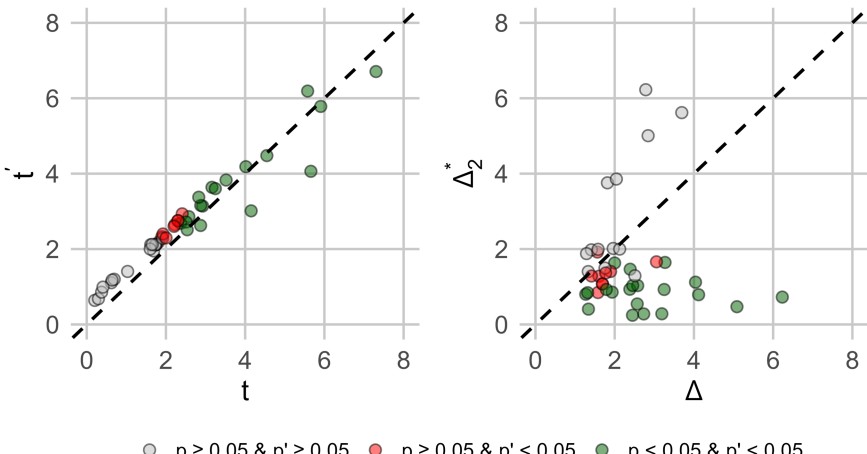

$$\text{Legend: } \quad \bigcirc \ p \geq 0.05 \ \& \ p' \geq 0.05 \qquad \bullet \ p \geq 0.05 \ \& \ p' < 0.05 \qquad \bullet \ p < 0.05 \ \& \ p' < 0.05$$

**Fig 4**. **Scatter plot illustrating the relationship between $t$ and $t'$ (left) and $\Delta$ and $\Delta_2^*$ (right) across the datasets examined in Experiment 3.** Each dot represents one dataset and dots are color coded based on the $p$-value with and without the maximum concordant observation.

results and contain concordant outliers, the strength of the t-statistic generally comes in spite of the magnitude of these outliers not as a product of it. In contrast, if we look at the ten cases where $\Delta_2^* > 2$, we see that none of these data show statistically significant test results either with or without the concordant observation present. This is not surprising since (by definition) $\Delta_2^*$ is only large when $\hat{\mu}$ is small. So, in summary, of the 50 datasets analyzed in this experiment, we could not find a single case where a statistically significant result could be attributed to an outlying sample by the definitions proposed in this paper.

### 3.4 Code availability

Code for the subsequent simulations can be found on Open Science Framework at https://osf.io/yfju9/. Simulations were conducted in R version 4.4.0 [28] and used dplyr for data manipulation [29], ggplot and patchwork for visualization[30,31], and BSDA and PairedData for data examples [26,27].

## 4 Discussion

The question we sought to investigate in this manuscript is whether it is possible for outliers, specifically concordant ones to *cause* false discoveries in one-sample or paired t-tests. While it is undoubtedly true under the definitions provided that outliers can lead to false discoveries, the conditions under which this occurs are surprisingly limited. Because the number of standard deviations above the mean that an outlying point maximally increases the t-statistic is shown to be inversely proportional to the effect size, outliers can only increase the t-statistic more than non-outlying values when the effect size is small-to-moderate (approximately $\hat{\mu}/\hat{\sigma} \leq 1/2$ for a $2\sigma$ outlier threshold). As smaller sample sizes require larger effect sizes to achieve statistical significance and the ability of single observations to increase the t-statistic is limited, we can generally rule out the possibility of outliers causing type I errors in small sample sizes ($n \leq 14$ for a $2\sigma$ outlier threshold). When considering smaller effect sizes, as the initial effect size approaches zero, the number of standard deviations away from the mean at which a point maximally increases the t-statistic approaches infinity. However, in practice, the extent to which such points can increase the t-statistic is limited and frequently insufficient to result in rejection of the null hypothesis. Using a 2-$\sigma$ outlier threshold, the highest observed p-value for which an outlier caused a significant result (i.e. both $\Delta \geq 2$ and $\Delta_2^* \geq 2$) was $p = 0.096$. Therefore, even in cases where outliers meet our criteria for causing a rejection of the null hypothesis, some initial indication of an effect must already be present in the original data.

To summarize, the following criteria must all be met for an outlier to cause a false discovery. First, the observed effect in the original data must be on the border of statistical significance. Based on our empirical findings, this alone probably occurs less than 5% of the time when the null hypothesis is true. Second, the dataset must contain a sufficient number of samples ($n > 15$) in order for effects near the significance threshold to be small enough to be positively affected by outliers. Finally, if the previous two criteria are met, the outlier still needs to be both: 1) concordant with the observed effect, and 2) not so great in magnitude that it starts to reduce the t-statistic. The narrowness of these conditions explains why the analysis of sample datasets did not find a single instance in which a significant effect could be attributed to an outlying sample, and justifies the titular claim that outliers typically cannot cause type I errors. These conditions also contrast starkly with the relatively broad conditions necessary for outliers to cause type II errors, which require only: 1) the original data exhibits a significant effect, and 2) the outlier is sufficiently large (in either direction) to nullify that effect. One part of this result that feels counter intuitive is the conclusion that outliers pose greater risk of causing false discoveries in large samples than small samples. Since a single corrupt observation constitutes a greater proportion of a small sample than a large one, it is natural to assume it would carry greater impact. This idea does not conflict with our findings. Examining Fig 2 shows that the number of cases where the p-value falls below the significance threshold as a result of the new observation (red dots) appears to at least as high in the smaller sample sizes ($n = 10, 25$). However, in the smaller sample sizes these cases tend to occur more frequently for smaller $\Delta$ values. Thus it is not the case that isolated observations cannot inflate the t-statistic in smaller samples, just that the ones doing so tend not to be far enough from the mean to be generally considered outliers.

Relating these findings to prior research, this is not the first study to show that outliers pose a greater risk for type II errors than type I errors in one-sample or paired t-tests [12,13,23,24]. However, few studies have focused specifically on the effects of concordant outliers. Our work builds naturally on the work of [25], who showed through simulation that increasing the magnitude of a concordant outlier will eventually result in increasing p-values and decreases to the null-hypothesis rejection rate. Building on this work, this study provides closed-form expressions for the point at which: 1) further increases to the outlier will begin to decrease the t-statistic ($\Delta_2^*$), and 2) outliers large enough to to decrease the t-statistic by their introduction ($\Delta_1^*$). In doing this, we have also established a clearer understanding of how the effect of concordant outliers on the t-statistic depends on the relationship between the outlier's magnitude, the sample size, and the effect size of the original data. To our knowledge, this is the first study to propose that outliers pose no risk of causing false-discoveries in one-sample or paired t-tests with small-sample designs.

This work also aligns well with existing theory within the robust statistics literature, which has made efforts to quantify that sensitivity of the t-statistic (and other measures) to corrupt data. One common measure of robustness that is closely related to the work in this paper is the breakdown point. The breakdown point of an estimator refers to the maximum percentage of data that can be corrupted before an estimator becomes unreliable (or exhibits arbitrarily large error) [32]. In the context of statistical inference the breakdown point can be separated into the *level breakdown point* (robustness of validity), which measures level of contamination needed to guarantee rejection of the null hypothesis, and the *power breakdown point* (robustness of efficiency), which measures the level of contamination needed to guarantee failure to reject the null hypothesis. Work that has examined these properties for the t-statistic have found it exhibits some robustness of validity, but little robustness of efficiency [17,33,34]. For the breakdown point specifically, the t-statistic has a power breakdown point of zero and a level breakdown point that is generally non-zero and proportional to $\mu/\sigma$ [33].

One important point to note about this analysis is that it makes no assumptions on the distribution from which the data is drawn. This means that this analysis, and the associated conclusions, are applicable regardless of whether the data is normally distributed or if violations of other standard assumptions such as heteroskedasticity or non-independence are present. As a result the conclusions we draw about whether or not an outlier is capable of causing an outlier for a given sample size, outlier magnitude, and effect size are not dependent on any of these distributional assumptions. So, for example, the claim that outliers cannot cause significant t-test results when $n < 15$ is true even when any or all of these assumptions are violated. Violation of these assumptions may yield inflated type I or type II error rates in the statistical

test, but it does not affect our conclusions about the ability of isolated outlying values to disproportionately impact a t-test towards a significant result. Note that the lack of distributional assumptions made in our analysis is a double edged sword. While it means that the conclusions in this paper are not reliant on properties of the underlying distribution, it also means that we cannot make claims about type I and type II error rates.

It is important to emphasize that these findings in no way diminish the concern that outliers, or more broadly corrupt data and the violation of distributional assumptions, may pose in one-sample or paired t-tests. While we make the case that isolated outliers cannot cause type I errors in small samples, an alternative interpretation of these results is that the erroneous data most likely to cause type I errors cannot be detected based on its deviation from the mean. Rather than diminishing the concern outliers present to 1-sample t-tests, this work reinforces the well established understanding that the primary concern outliers present in this context is diminished efficiency and reduced statistical power. Thus, in cases of known contamination or non-normality alternative robust or non-parametric methods may still be appropriate. For this, there are a range of robust methods that attempt to maintain efficiency close to that of parametric methods while reducing sensitivity to violations of model assumptions. These include strategies based on trimmed means [18], bootstrapping methods [19,20], and M-estimators [18,21,22]. Non-parametric methods can also provide a compelling option in some cases, but are not advisable for every application [35–37]. This work is not meant to discourage the use of such alternatives, but to facilitate proper interpretation of the t-test in cases where it is used with outliers present.

Perhaps the most restrictive limitation imposed by our analysis is that only isolated outliers are considered. Some study of the effects of multiple concordant outliers can be found in the analysis presented in [34], which examines the effects of asymmetric contamination and finds that large contaminator rates (15%) can lead to significantly inflated type I error rates. To better understand the effects of multiple outliers, this framework could be used to test the iterative addition of points at $\Delta_2^*$ and assess what the minimal number of optimally placed corrupt observations is necessary to produce a significant result in different scenarios. However, considering the joint impact of multiple concordant outliers remains a topic for future work. One final point to note regarding the application of these ideas to other settings is that the framework used here is an inversion of how outliers are usually characterized. Very often outliers are considered as part of the original sample and considered for removal, whereas our framework considers the addition of a possibly outlying data point to a *clean* original sample. So when we say, for example, that $\Delta_2^*$ is the number of standard deviations above the mean at which a new observation maximally increases the test statistic, the mean and standard deviations values referred to here are both measured without the new observation. As a result, these formulas do not apply exactly to the context of outlier removal, but can easily be implemented in that context by re-measuring these terms sans outlier, as shown in Experiment 3. As a final note, it is worth mentioning that the $\sigma$-based outlier criteria used in this paper are chosen for convenience, and are generally not recommended as a method for identifying outliers [11,38].

## 5 Conclusion

This study investigates how outlying values influence the results of one-sample or paired t-tests, specifically in cases where the outlier is concordant with the direction of the effect observed in the rest of the data. We show that, even in these cases, outliers frequently decrease rather than increase the test statistic. To formalize this, we introduced bounds that provide upper limits on how large an outlier can be before it begins to reduce the test statistic. These bounds show that for small sample sizes ($n < 15$), adding an outlier more than two standard deviations above the mean cannot produce a significant result that could not already be achieved by a non-outlier. In larger samples, outliers can cause significant results that non-outlying values cannot, but only under relatively constrained circumstances.

## Supporting information

**S1 Appendix A. Basic derivations.**
(PDF)

**S2 Appendix B. Proof of Theorem 1.**
(PDF)

**S3 Appendix C. Proof of Theorem 2.**
(PDF)

**S4 Appendix D. Non-normal simulations.**
(DOCX)

## Author contributions

**Conceptualization:** Alan Wisler.

**Formal analysis:** Alan Wisler.

**Investigation:** Alan Wisler.

**Methodology:** Alan Wisler.

**Project administration:** Alan Wisler.

**Validation:** Alan Wisler.

**Visualization:** Alan Wisler.

**Writing – original draft:** Alan Wisler.

**Writing – review & editing:** Alan Wisler.

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
