## [Decision Letter · Decision Letter 0]

3 Nov 2025

PONE-D-25-53826Outliers (typically) cannot cause type I errors in one-sample / paired t-testsPLOS ONE

Dear Dr. Wisler,

Thank you for submitting your manuscript to PLOS ONE. After careful consideration, we feel that it has merit but does not fully meet PLOS ONE’s publication criteria as it currently stands. Therefore, we invite you to submit a revised version of the manuscript that addresses the points raised during the review process.

We look forward to receiving your revised manuscript.

Kind regards,

Abhik Ghosh

Academic Editor

PLOS ONE

Journal Requirements:

Reviewers' comments:

Reviewer's Responses to Questions

**Comments to the Author**

1. Is the manuscript technically sound, and do the data support the conclusions?

Reviewer #1: Yes

Reviewer #2: Partly

Reviewer #3: Yes

Reviewer #4: Yes

2. Has the statistical analysis been performed appropriately and rigorously?

Reviewer #1: Yes

Reviewer #2: Yes

Reviewer #3: Yes

Reviewer #4: Yes

3. Have the authors made all data underlying the findings in their manuscript fully available?

Reviewer #1: Yes

Reviewer #2: Yes

Reviewer #3: Yes

Reviewer #4: Yes

4. Is the manuscript presented in an intelligible fashion and written in standard English?

Reviewer #1: Yes

Reviewer #2: Yes

Reviewer #3: Yes

Reviewer #4: Yes

5. Review Comments to the Author

Reviewer #1: This manuscript tackles a pervasive and often misunderstood problem in statistical inference with a commendable blend of theoretical derivation and simulation. The findings are potentially significant for guiding empirical research. However, the work requires major revision to enhance its clarity, robustness, and practical applicability. The assumptions are quite restrictive, and the presentation of the core theorems could be made more accessible to the broader target audience of applied researchers.

1. The assumption of unit variance (σ^x2=1σ^x2=1), while mathematically convenient for derivation, limits the immediate interpretability for applied researchers. The manuscript should include a more detailed discussion or a supplementary section on how to apply these bounds to real-world data where this assumption does not hold, perhaps with a worked example.

2. The definition of an outlier "causing" rejection is logically sound but operationally challenging. The manuscript should discuss the practical implications of this definition, especially in scenarios where multiple outliers might be present, and how a practitioner might distinguish between an outlier-caused discovery and a genuine one.

3. The simulation is well-designed but could be strengthened by testing the robustness of the derived bounds against violations of normality. Including data generated from heavy-tailed distributions (e.g., t-distribution) would demonstrate the generalizability of the findings beyond the ideal Gaussian case assumed in the model.

4. The connection between the theoretical bounds and practical outlier detection rules (e.g., 2-σ or 3-σ) is a key contribution. However, the discussion would be enriched by engaging with more modern, robust statistical methods. The authors are strongly encouraged to review Robust Statistical Methods for Handling Outliers in High-Dimensional Data, Journal of the American Statistical Association; A robust identification method for stochastic nonlinear parameter varying systems, Mathematical Modelling and Control; Extended state observer based fractional order sliding mode control for steer-by-wire systems, IET Control Theory and Applications. These works demonstrate advanced techniques for handling uncertainties and outliers in dynamic systems, which could provide a broader context for the statistical problem addressed here and inspire more robust interpretations.

5. The manuscript concludes that the risk is "negligible" in many situations. This is a strong claim that should be tempered. A more nuanced conclusion, acknowledging the specific and narrow conditions under which outliers can cause type I errors, would be more accurate and helpful for the reader.

6. The abstract and introduction effectively set up the problem, but the "Methods and Results" section is dense and could benefit from additional explanatory text to guide the reader through the logical flow of the derivations, making the paper more accessible to those less versed in mathematical statistics.

Reviewer #2: The manuscript titled “Outliers (typically) cannot cause Type I errors in one-sample/paired t-tests” presents a technically rigorous and mathematically sound analysis of how outliers influence Type I error rates in t-tests. The work is well motivated and addresses an important and long-debated topic in applied statistics. The author combines theoretical derivations with simulation evidence to demonstrate that, under controlled assumptions, outliers rarely inflate Type I errors. The findings are internally consistent and supported by the presented data.

That said, the scope of the contribution is somewhat limited, and the paper would benefit from several substantive and editorial improvements before it can meet the standards of PLOS ONE. My detailed observations and suggestions are as follows:

1. Technical Strengths

The mathematical framework and derivations are accurate, logically consistent, and presented in a reproducible manner.

The Monte Carlo simulations are appropriate in design and provide convincing numerical support for the theoretical results.

The results contribute to the broader discussion of robustness in classical parametric testing, clarifying misconceptions about the role of outliers in Type I error inflation.

2. Limitations and Areas for Improvement

a. Scope and Novelty:

While the results are interesting, the contribution is incremental rather than groundbreaking. Several prior studies (e.g., Zimmerman, 1994; Derrick et al., 2017; Wilcox, 2020) have reached similar conclusions through simulation-based approaches. The current work’s novelty lies mainly in providing theoretical bounds (Δ₁, Δ₂), which could be valuable but do not represent a major conceptual advance for a multidisciplinary audience.

b. Generalizability of Findings:

The analysis is confined to data simulated from normal distributions with fixed variance and controlled outlier contamination. To strengthen generalizability, consider:

Including non-normal or heavy-tailed distributions (e.g., t-distribution, log-normal) to demonstrate robustness.

Discussing whether similar results hold under heteroscedasticity or when data violate independence assumptions.

c. Empirical Relevance:

All results are based solely on simulations. Including a short applied example—even using an open dataset from behavioral or biomedical research—would substantially enhance the practical relevance and reader engagement. This could also highlight real-world implications of the derived bounds.

d. Presentation and Clarity:

The manuscript is generally well written but would benefit from professional language editing to correct typographical errors, long sentences, and redundant phrasing.

Figures are informative, but some could use clearer legends and axis labels.

Ensure consistent notation for mathematical symbols and parameters throughout (e.g., Δ₁, Δ₂, μ̂, σ̂).

The abstract should explicitly summarize the methodological contribution and practical implications rather than focusing solely on background.

e. Literature Review:

The literature review is appropriate but can be enhanced by citing and contrasting recent robust statistical inference studies (2022–2024), including methods involving M-estimators, bootstrap-based robust t-tests, or influence function analyses. This would contextualize the current findings within the evolving field of robust statistics.

3. Data Availability and Reproducibility

The author appropriately provides simulation code and data availability details via an OSF repository, aligning with PLOS ONE’s open-data policy. However, please ensure the exact repository link is included in the Data Availability Statement and that sufficient details (software version, random seeds, replication size) are specified for full reproducibility.

4. Overall Evaluation

The study is methodologically solid, mathematically correct, and clearly motivated. However, the breadth of contribution and applied impact are limited, making the work more appropriate for a specialized statistical methods journal than a general multidisciplinary platform like PLOS ONE.

If the author expands the simulation scope, improves clarity, and adds an applied example, the manuscript could become suitable for reconsideration.

Summary Recommendation

Based on the current form, I recommend rejection due to limited novelty and restricted scope for the broad readership of PLOS ONE. However, the work is of good technical quality and may merit publication in a specialized statistical journal after revision and expansion.

Reviewer #3: Overall Assessment

The paper “Outliers (typically) cannot cause type I errors in one-sample/paired t-tests” is technically sound and well written. The introduced mathematical bounds are both simple and ingenious, and the results are presented clearly. However, several aspects of the manuscript could be improved to enhance clarity, completeness, and presentation.

Major Concerns

1. Introduction – Insufficient bibliography

The introduction would benefit from a more comprehensive review of related work. For instance, in line 49, additional examples or references could be added.

2. Introduction – Missing paper outline

The manuscript lacks a clear outline of its structure. A brief description of the organization (e.g., “Section 2 introduces…, Section 3 presents…”) would guide readers and improve readability.

3. Lack of empirical validation

The article would benefit from including an analysis of a real dataset where the proposed bounds can be observed in practice.

Minor Concerns

4. Figure 1 – Axis readability

The axes in Figure 1 are too small.

5. Discussion – Use of subjective adjectives

Avoid using subjective or qualitative adjectives such as “small-to-moderate,” “large,” “not too large,” “good,” and “bad” when discussing the results. More precise or quantitative descriptions are recommended.

Reviewer #4: In this paper, the authors investigate whether a single outlier can cause the rejection of the null hypothesis in a one-sample t-test. The main contributions are two theorems that characterize the magnitude of potentially harmful outliers. First, the authors show that outliers above a certain magnitude can only decrease the test statistic and therefore cannot cause type I errors. Second, they derive an explicit expression for the outlier magnitude that maximizes the effect on the test statistic. The results are further discussed and illustrated with Monte Carlo simulations.

The topic of outlier handling is timely and important. To the best of my knowledge, the results presented in this paper are correct, novel, and relevant both in theory and in practice. The paper is well written, the proofs are transparent and the simulation results convincing. Overall, I found this to be a very interesting and enjoyable read, and I recommend the paper for publication.

I have, however, a few comments and suggestions. Some of them may go beyond the intended scope of the paper, so I leave it to the author’s discretion which ones to incorporate.

1. The approach taken in the paper seems closely related to ideas from robust statistics, in particular to (empirical) influence functions. The problem considered here could likely be embedded in that framework by viewing the t-statistic as an estimator of a standardized mean difference and then studying its sensitivity to point-mass contamination. Classic treatments of this perspective can be found in [1]. Referring to this line of work would help situate the problem and show that the present result is consistent with a more general robustness perspective. That said, I am not sure how far it is worth pursuing this in the current paper — in the worst case, it may overcomplicate an otherwise clean argument. Still, I would recommend at least pointing out that this connection exists.

2. Some results are stated somewhat implicitly. For example, I understand that Δ_1^* > Δ_2^*. Is this correct If so, it would be helpful to state and prove it explicitly. Also, is the inequality strict for some (or all) sample sizes and sample means? Based on Theorem 2, it should also be possible to give an explicit expression for the maximum difference t' - t.

3. It might help some readers to add a plot of t' (and the original t, as a constant) as functions of Δ for fixed μ^ and n. Highlighting Δ_1^* and Δ_2^* in this plot would make the results more transparent.

4. Intuitively, one might expect smaller sample sizes to lead to tests that are more sensitive to outliers. Here, the paper shows a somewhat counterintuitive phenomenon: extreme outliers can cause type I errors only for sufficiently large sample sizes. At the same time, the area in the μ^-Δ plane in which outliers cause type I errors does seem to decrease as the sample size increases. Maybe the author could expand the discussion on these effects.

5. A practically very relevant question is: Given a significance level and an observed t-value that leads to a rejection of the null hypothesis, is it possible that this rejection was caused by an outlier in the data? My impression is that the results in this paper can be used to answer this question.

6. As the author notes, extending the analysis to multiple outliers is a natural next step. The analogue of comment 5 would then be: How many data points would need to be outliers for the rejection of the null hypothesis to be possibly caused to them?

7. Minor issues:

- In the derivation of the “New Sample Variance,” the cross term in the third line is incorrect. The final result is still correct because the term is zero, but the intermediate line should be fixed.

- In line 156, a word seems to be missing: “the data cannot [increase?] the t-statistic.”

[1] Hampel, F. R., Ronchetti, E. M., Rousseeuw, P. J., & Stahel, W. A. (1986), Robust Statistics: The Approach Based on Influence Functions (Wiley)

6. PLOS authors have the option to publish the peer review history of their article (what does this mean?). If published, this will include your full peer review and any attached files.

Reviewer #1: No

Reviewer #2: No

Reviewer #3: No

Reviewer #4: No

---

## [Author Response · Author response to Decision Letter 1]

8 Dec 2025

Formatted responses in "Response to Reviewer" letter.

Reviewer #1

This manuscript tackles a pervasive and often misunderstood problem in statistical inference with a commendable blend of theoretical derivation and simulation. The findings are potentially significant for guiding empirical research. However, the work requires major revision to enhance its clarity, robustness, and practical applicability. The assumptions are quite restrictive, and the presentation of the core theorems could be made more accessible to the broader target audience of applied researchers.

Reviewer 1:

1. The assumption of unit variance (σ^x2=1σ^x2=1), while mathematically convenient for derivation, limits the immediate interpretability for applied researchers. The manuscript should include a more detailed discussion or a supplementary section on how to apply these bounds to real-world data where this assumption does not hold, perhaps with a worked example.

Response: Thank you for pointing this out, the original submission did not adequately explain why this assumption is not limiting. Since any data can be rescaled to meet these conditions, they do not actually limit the applicability of these methods. To clarify this, we have reframed these as “conditions” rather than “assumptions” and provided additional explanation for why they do not restrict the analysis. Please see lines 113-121 for the revised text.

Reviewer 1:

2. The definition of an outlier "causing" rejection is logically sound but operationally challenging. The manuscript should discuss the practical implications of this definition, especially in scenarios where multiple outliers might be present, and how a practitioner might distinguish between an outlier-caused discovery and a genuine one.

Response: This is an excellent point which has been echoed by some of the other reviewers. To address this, we have added a third experiment that walks readers through how this analysis can be applied to real-world datasets.

Reviewer 1:

3. The simulation is well-designed but could be strengthened by testing the robustness of the derived bounds against violations of normality. Including data generated from heavy-tailed distributions (e.g., t-distribution) would demonstrate the generalizability of the findings beyond the ideal Gaussian case assumed in the model.

Response: Since the effect of a new observation on the t-statistic in this framework can be quantified entirely by n,\Delta,\ &\ \hat{\mu}, it does not actually matter which distribution is used here. To demonstrate this, in our revised submission we have included supplemental material that shows identical results for data from a 1) t-distribution and 2) a deterministic binary sequence. We have also provided additional text to clarify this point in our methods (lines 238-244) and discussion (lines 271-286)

Reviewer 1:

4. The connection between the theoretical bounds and practical outlier detection rules (e.g., 2-σ or 3-σ) is a key contribution. However, the discussion would be enriched by engaging with more modern, robust statistical methods. The authors are strongly encouraged to review Robust Statistical Methods for Handling Outliers in High-Dimensional Data, Journal of the American Statistical Association; A robust identification method for stochastic nonlinear parameter varying systems, Mathematical Modelling and Control; Extended state observer based fractional order sliding mode control for steer-by-wire systems, IET Control Theory and Applications. These works demonstrate advanced techniques for handling uncertainties and outliers in dynamic systems, which could provide a broader context for the statistical problem addressed here and inspire more robust interpretations.

Response: Thank you for these suggestions. We were unable to find the first reference and had difficulty placing the third reference within our review. However, we have added the second reference you mentioned (Morato et al. 2021) and two additional papers conceptually similar to the JASA article we were unable to find in lines 19-21. If you can provide a DOI link for the first paper we would be happy to include it as well.

Reviewer 1:

5. The manuscript concludes that the risk is "negligible" in many situations. This is a strong claim that should be tempered. A more nuanced conclusion, acknowledging the specific and narrow conditions under which outliers can cause type I errors, would be more accurate and helpful for the reader.

Response: We have revised this sentence in the abstract to instead say “minimal” and include an additional caveat of “isolated outliers”. While this is still strong language, we believe it is justified by the results and a more concrete and nuanced description of the result is presented immediately before this claim in the abstract.

Reviewer 1:

6. The abstract and introduction effectively set up the problem, but the "Methods and Results" section is dense and could benefit from additional explanatory text to guide the reader through the logical flow of the derivations, making the paper more accessible to those less versed in mathematical statistics.

Response: Thank you for pointing this out. We have substantially revised the methods and results to improve clarity for a general audience.

Reviewer #2:

The manuscript titled “Outliers (typically) cannot cause Type I errors in one-sample/paired t-tests” presents a technically rigorous and mathematically sound analysis of how outliers influence Type I error rates in t-tests. The work is well motivated and addresses an important and long-debated topic in applied statistics. The author combines theoretical derivations with simulation evidence to demonstrate that, under controlled assumptions, outliers rarely inflate Type I errors. The findings are internally consistent and supported by the presented data.

That said, the scope of the contribution is somewhat limited, and the paper would benefit from several substantive and editorial improvements before it can meet the standards of PLOS ONE. My detailed observations and suggestions are as follows:

1. Technical Strengths

The mathematical framework and derivations are accurate, logically consistent, and presented in a reproducible manner.

The Monte Carlo simulations are appropriate in design and provide convincing numerical support for the theoretical results.

The results contribute to the broader discussion of robustness in classical parametric testing, clarifying misconceptions about the role of outliers in Type I error inflation.

Reviewer 2:

2. Limitations and Areas for Improvement

a. Scope and Novelty:

While the results are interesting, the contribution is incremental rather than groundbreaking. Several prior studies (e.g., Zimmerman, 1994; Derrick et al., 2017; Wilcox, 2020) have reached similar conclusions through simulation-based approaches. The current work’s novelty lies mainly in providing theoretical bounds (Δ₁, Δ₂), which could be valuable but do not represent a major conceptual advance for a multidisciplinary audience.

Response: This research was partially motivated by an observed general lack of understanding of the limited risk outliers pose in causing type-I error in one sample t-tests; and a lack of existing research that clearly lays out this fact. While the reference prior studies clearly show that outliers affect type-II error rate more than type-I error rate, we did not feel they adequately communicated the narrowness of circumstances necessary for outliers to cause type I errors or their inability to cause type I errors in small sample regimes.

Reviewer 2:

b. Generalizability of Findings:

The analysis is confined to data simulated from normal distributions with fixed variance and controlled outlier contamination. To strengthen generalizability, consider:

Including non-normal or heavy-tailed distributions (e.g., t-distribution, log-normal) to demonstrate robustness.

Discussing whether similar results hold under heteroscedasticity or when data violate independence assumptions.

Response: This paper makes no distributional assumptions. Thus, if a sample (of size n) produces a specific sample mean and variance it will bare the same sensitivity to the new observation regardless of whether the process generating the data is non-normal, or exhibits heteroscedasticity or non-normality. To demonstrate this, in our revised submission we have included supplemental material that shows identical results for data from a 1) t-distribution and 2) a deterministic binary sequence. We have also provided additional text to clarify this point in our methods (lines 238-244) and discussion (lines 271-286)

Reviewer 2:

c. Empirical Relevance:

All results are based solely on simulations. Including a short applied example—even using an open dataset from behavioral or biomedical research—would substantially enhance the practical relevance and reader engagement. This could also highlight real-world implications of the derived bounds.

Response: Thank you for the suggestion. We have added a third experiment that examines a diverse set of sample paired datasets.

Reviewer 2:

d. Presentation and Clarity:

The manuscript is generally well written but would benefit from professional language editing to correct typographical errors, long sentences, and redundant phrasing.

Response: Thank you for the suggestion. We have given the manuscript a thorough review and corrected all the typographical errors we could identify. We also reviewed for redundant phrasing and found some overuse of certain phrases (“So”, “however”, “Thus”, “Note that”), and tried to used alternative phrasing where these words appeared too frequently in succession.

Reviewer 2:

Figures are informative, but some could use clearer legends and axis labels.

Response: We have revised the figures, particularly figure 1 (now figure 2 in the resubmission), for improved clarity. We have also tried to expand the captions to more comprehensively describe the figures.

Reviewer 2:

Ensure consistent notation for mathematical symbols and parameters throughout (e.g., Δ₁, Δ₂, μ̂, σ̂).

Response: We have reviewed the notation in the abstract and corrected some minor notational inconsistencies. Please let us know if you spot any that we missed.

Reviewer 2:

The abstract should explicitly summarize the methodological contribution and practical implications rather than focusing solely on background.

Response: We have revised the abstract for clarity. A majority of the revised abstract, including everything after “Towards this end,…” reflects the specific contributions of this manuscript.

Reviewer 2:

e. Literature Review:

The literature review is appropriate but can be enhanced by citing and contrasting recent robust statistical inference studies (2022–2024), including methods involving M-estimators, bootstrap-based robust t-tests, or influence function analyses. This would contextualize the current findings within the evolving field of robust statistics.

Response: Thank you for this suggestion. We have significantly expanded the references to more than double the number contained in the original submission. Much of these new references focus on recent work in robust statistics.

Reviewer 2:

3. Data Availability and Reproducibility

The author appropriately provides simulation code and data availability details via an OSF repository, aligning with PLOS ONE’s open-data policy. However, please ensure the exact repository link is included in the Data Availability Statement and that sufficient details (software version, random seeds, replication size) are specified for full reproducibility.

Response: Thank you for the suggestion. We have added a code availability statement to the end of the methods which include: the OSF link as well citations and version information for R and all packages used in the experiment. The OSF link has also been updated to include code for the additional experiment, and the provided code has been tested on multiple devices. All experiments also use predefined seeds for replication. If there is anything else we can provide to enhance reproducibility please let us know.

Reviewer 2:

4. Overall Evaluation

The study is methodologically solid, mathematically correct, and clearly motivated. However, the breadth of contribution and applied impact are limited, making the work more appropriate for a specialized statistical methods journal than a general multidisciplinary platform like PLOS ONE.

If the author expands the simulation scope, improves clarity, and adds an applied example, the manuscript could become suitable for reconsideration.

Summary Recommendation

Based on the current form, I recommend rejection due to limited novelty and restricted scope for the broad readership of PLOS ONE. However, the work is of good technical quality and may merit publication in a specialized statistical journal after revision and expansion.

Response: We appreciate your perspective and hope the extensive updates we have made to the paper in these revisions are sufficient to change your opinion.

Reviewer #3:

Overall Assessment

The paper “Outliers (typically) cannot cause type I errors in one-sample/paired t-tests” is technically sound and well written. The introduced mathematical bounds are both simple and ingenious, and the results are presented clearly. However, several aspects of the manuscript could be improved to enhance clarity, completeness, and presentation.

Reviewer 3:

Major Concerns

1. Introduction – Insufficient bibliography

The introduction would benefit from a more comprehensive review of related work. For instance, in line 49, additional examples or references could be added.

Response: Thank you for the suggestion. In line with this comment and those of the other reviewers, we have substantially expanded the review of related work, particularly emphasizing work in robust statistics that was not reviewed in the original submission.

Reviewer 3:

2. Introduction – Missing paper outline

The manuscript lacks a clear outline of its structure. A brief description of the organization (e.g., “Section 2 introduces…, Section 3 presents…”) would guide readers and improve readability.

Response: Thank you for the suggestion, we have added an outline in lines 77-87.

Reviewer 3:

3. Lack of empirical validation

The article would benefit from including an analysis of a real dataset where the proposed bounds can be observed in practice.

Response: Thank you for the suggestion. We have added a third experiment that examines a diverse set of sample paired datasets to help show readers how these bounds work in practical settings.

Reviewer 3:

Minor Concerns

4. Figure 1 – Axis readability

The axes in Figure 1 are too small.

Response: We have revised the experiment 1 figure for improved readability.

Reviewer 3:

5. Discussion – Use of subjective adjectives

Avoid using subjective or qualitative adjectives such as “small-to-moderate,” “large,” “not too large,” “good,” and “bad” when discussing the results. More precise or quantitative descriptions are recommended.

Response: Thank you for the suggestion. While we believe this subjective terminology is helpful in relating these ideas to a general audience, we agree they should not be used without precise quantitative grounding. To remedy this, we have reviewed the paper and tried to support every use of subjective language with precise quantitative grounding. For example “small-to-moderate (approximately \hat{\mu}/\hat{\sigma}\le1/2...” uses subjective language but then precisely states what numerical values this language refers to. Please let us know if any cases remain too imprecise.

Reviewer #4:

In this paper, the authors investigate whether a single outlier can cause the rejection of the null hypothesis in a one-sample t-test. The main contributions are two theorems that characterize the magnitude of potentially harmful outliers. First, the authors show that outliers above a certain magnitude can only decrease the test statistic and therefore cannot cause type I errors. Second, they derive an explicit expression for the outlier magnitude that maximizes the effect on the test statistic. The results are further discussed and illustrated with Monte Carlo s

---

## [Decision Letter · Decision Letter 1]

12 Jan 2026

Outliers (typically) cannot cause type I errors in one-sample / paired t-tests

PONE-D-25-53826R1

Dear Dr. Wisler,

We’re pleased to inform you that your manuscript has been judged scientifically suitable for publication and will be formally accepted for publication once it meets all outstanding technical requirements.

Kind regards,

Abhik Ghosh

Academic Editor

PLOS One

Additional Editor Comments (optional):

Reviewers' comments:

Reviewer's Responses to Questions

**Comments to the Author**

1. If the authors have adequately addressed your comments raised in a previous round of review and you feel that this manuscript is now acceptable for publication, you may indicate that here to bypass the “Comments to the Author” section, enter your conflict of interest statement in the “Confidential to Editor” section, and submit your "Accept" recommendation.

Reviewer #1: All comments have been addressed

Reviewer #2: All comments have been addressed

Reviewer #3: All comments have been addressed

Reviewer #4: All comments have been addressed

2. Is the manuscript technically sound, and do the data support the conclusions?

Reviewer #1: Yes

Reviewer #2: Yes

Reviewer #3: Yes

Reviewer #4: Yes

3. Has the statistical analysis been performed appropriately and rigorously?

Reviewer #1: Yes

Reviewer #2: Yes

Reviewer #3: Yes

Reviewer #4: Yes

4. Have the authors made all data underlying the findings in their manuscript fully available?

Reviewer #1: Yes

Reviewer #2: Yes

Reviewer #3: Yes

Reviewer #4: Yes

5. Is the manuscript presented in an intelligible fashion and written in standard English?

Reviewer #1: Yes

Reviewer #2: Yes

Reviewer #3: Yes

Reviewer #4: Yes

6. Review Comments to the Author

Reviewer #1: (No Response)

Reviewer #2: Strengths

1. Strong theoretical contribution with clear derivations and formal proofs.

2. Well-designed simulations that directly validate theoretical results.

3. High relevance to applied researchers who routinely face decisions about outlier handling.

4. Excellent reproducibility, with simulation code openly available.

Minor Concerns / Suggestions

1. Title tone

The word “typically” in the title is appropriate but may be perceived as informal or ambiguous. The author may consider briefly clarifying its meaning in the introduction.

2. Practical guidance

While the theoretical insights are strong, the manuscript could benefit from a short subsection translating results into practical recommendations for applied researchers (e.g., when outlier removal is unlikely to inflate Type I error).

3. Discussion length

The Discussion section is thorough but could be slightly condensed to improve readability without loss of content.

4. Terminology consistency

The interchangeable use of “one-sample” and “paired” t-tests is justified but may briefly confuse non-technical readers; a reminder statement may help.

These are minor issues and do not affect the scientific validity of the work.

Reviewer #3: The comments for the paper titled "Outliers (typically) cannot cause type I errors in one-sample / paired t-tests" were properly addresed.

Reviewer #4: I appreciate the revised manuscript and the detailed responses to the reviewers’ comments. Overall, the paper has been significantly improved. The presentation is clearer, the generality of the results is made more explicit, the work is better positioned in the wider context of robust statistics, and the expanded bibliography is much more comprehensive.

Three minor comments:

- Figure 1 is a useful addition to the paper and provides a lot of information. It may in fact provide too much information, to the extend that it appears visually overloaded. I would suggest simplifying and decluttering the figure. For example, the number of curves could be reduced, the visualization of \Delta_1 and \Delta_2 could be simplified, and the derivative shown in the rightmost panel seems unnecessary for the discussion of the results. I believe that a simpler plot, such as the one in the attached file, would be sufficient to illustrate \Delta_1 and \Delta_2. The dependence of the latter on \mu^ and n could be shown in a separate figure, if at all.

- On rereading the paper, I agree with the comments that the language could be more objective and precise in places, even in the revised version. For example, in the abstract, it is unclear whether the term "minimal" is used informally (to mean small or negligible) or in the formal sense of "attaining a minimum". Similarly, it is not clear what is meant by describing the experiments as "unique." More generally, words such as "actually" are rarely necessary in formal writing. One concrete suggestion: the phrase "we impose the following two conditions, that do not actually restrict the generality of the analysis" could be replaced with "without loss of generality, we assume that..."

- I found the extended discussion section very helpful. One point addressed there that could be highlighted earlier is that the paper focuses specifically on whether outliers (extreme values) in the data can lead to false positives. It does not address the more general question of the sensitivity of the t-test to other forms of model mismatch or data contamination. Clarifying this distinction earlier in the paper may help avoid confusion.

7. PLOS authors have the option to publish the peer review history of their article (what does this mean?). If published, this will include your full peer review and any attached files.

Reviewer #1: No

Reviewer #2: No

Reviewer #3: No

Reviewer #4: No

---

## [Editor Report · Acceptance letter]

PONE-D-25-53826R1

PLOS One

Dear Dr. Wisler,

I'm pleased to inform you that your manuscript has been deemed suitable for publication in PLOS One. Congratulations! Your manuscript is now being handed over to our production team.

Kind regards,

on behalf of

Dr. Abhik Ghosh

Academic Editor

PLOS One